# Rising infrastructure inequalities accompany urbanization and economic development

Bhartendu Pandey[1,2] ✉, Christa Brelsford [3] & Karen C. Seto [2]

Impending global urban population growth is expected to occur with considerable infrastructure expansion. However, our understanding of attendant infrastructure inequalities is limited, highlighting a critical knowledge gap in the sustainable development implications of urbanization. Using satellite data from 2000 to 2019, we examine country-level population-adjusted biases in infrastructure distribution within and between regions of varying urbanization levels and derive four key findings. First, we find long-run positive associations between infrastructure inequalities and both urbanization and economic development. Second, our estimates highlight increasing infrastructure inequalities across most of the countries examined. Third, we find greater future infrastructure inequality increases in the global south, where inequalities will rise more in countries with substantial urban primacy. Fourth, we find that infrastructure inequality may evolve differently than economic inequalities. Overall, advancing sustainable development vis-à-vis urbanization and economic development will require intentional infrastructure planning for spatial equity.

How will urban infrastructure inequalities change with future urbanization? Recent related academic literature and policy discourse emphasize the challenges and opportunities that accompany urban demographic growth and attendant infrastructure expansion[1,2]. However, relatively little is known about infrastructural inequalities. This knowledge gap limits our understanding of whether and how urban infrastructural inequalities can constrain universal access to urban services, lock in social and economic inequalities[3], affect human health and wellbeing[4], interface with urban climate mitigation and adaptation challenges and opportunities[5,6], and ultimately influence progress towards sustainable development[7,8]. Understanding these multi-dimensional implications, each with substantial science and policy relevance, requires advancing our understanding of the characteristics and dynamics of urban infrastructural inequalities.

Urban areas are the epicenters of social diversity, economic efficiency, and opportunities[9,10], and simultaneously multi-dimensional inequalities[11–13]. On the latter, a substantial science and public policy focus is on economic inequalities, i.e., how does the distribution of

wages, income, or wealth vary across individuals and households, social groups, and neighborhoods within and across urban areas[14–16]. Beyond inequalities in economic outcomes, considerable disparities in the urban built environment exist and have been the focus of many recent studies[17–21], with some emphasizing urban infrastructure gaps[22,23]. Other relevant studies highlight the inequality implications of infrastructure[24–26]. Still, variations in urban infrastructural inequalities across contexts and how these inequalities change over time remain unclear.

Understanding urban infrastructural inequalities faces at least two epistemic challenges. First, urban inequalities are generally studied assuming an urban and rural dichotomy, which is often challenged due to extensive transboundary urban-rural linkages[27] and considering urbanization as a multi-dimensional process within which typically urban infrastructural amenities and associated lifestyles can disperse from urban areas to rural areas[28,29]. Second, infrastructure is an expansive term with multiple physical and social components[30,31], each subject to unequal distributions. Its diverse characteristic complicates

[1]Geospatial Science and Human Security Division, National Security Sciences Directorate, Oak Ridge National Laboratory, TN Oak Ridge, USA. [2]Yale School of the Environment, Yale University, New Haven, CT, USA. [3]Analytics, Intelligence, and Technology Division, Los Alamos National Laboratory, Los Alamos, NM, USA. ✉e-mail: pandeyb1@ornl.gov

our scientific understanding of inequalities' patterns, drivers, and socio-environmental implications—as the characteristics of inequalities along one infrastructural dimension can differ from that of other dimensions. Based on results from existing research, the use of satellite remote sensing-derived nighttime lights (NTL) can help address these challenges.

Previous research documented varied socio-economic applications of NTL[32] to complement ground surveys, and highlighted their importance for the measurement of human-related progress towards sustainable development[33]. Measuring regional inequalities from NTL is an emerging application area[34–37], but it exclusively examines NTL as a proxy measure of economic output (in monetary units). Near-surface NTL emissions are related primarily to energy infrastructure (such as buildings and outdoor street lighting) and nighttime (such as vehicular) activities, which in some ways determine regional economic output[38]. Accordingly, NTL distributions can specify the level of inequalities associated with aggregate infrastructure availability when analyzed with population distributions. The wall-to-wall spatial coverage and annual frequency coverage of NTL further allow examining how infrastructure inequalities vary across consistent spatial and temporal scales.

Here, we study the structure and dynamics of urban infrastructure inequalities—to what extent are infrastructure distributions preferentially allocated within and between regions of varying degrees of urbanization—using a measure suited for quantifying infrastructure inequalities from NTL[39]. In effect, we examine the complete rural-to-urban spectrum specified by variations in infrastructure availability within and across regions, conceptualizing infrastructure as an urbanization dimension[40–43]. In doing so, we implement the degree of urbanization proposition recently endorsed by the United Nations[44]. Additionally, we account for the diffusive nature of urban infrastructure and amenities extending from large urban areas to small settlements, aligning with the notion of planetary urbanization[29]. In our measurement approach, we use a default lattice grid at 0.5° spatial resolution, where each grid cell represents a region with a specific degree of urbanization. With this approach, our experimental design foundationally delves into the criticality of addressing inequalities within urban areas towards sustainable development—as put forward in the sustainable development goal (SDG) 11—as well as regionally considering urbanization as a spatially extensive process. Our study addresses four key research questions:

(1) How do urbanization—urban share of the total population—and economic development associate with infrastructural inequalities?

(2) How have infrastructural inequalities changed in the past twenty years (2000–2019)? Are these changes predictable?

(3) How will infrastructural inequalities change in the future under different urbanization and economic development scenarios?

(4) How do infrastructure inequality forecasts compare with existing economic inequality forecasts?

Describing overall (national) inequality with regional infrastructure distributions necessitates a spatial decomposition into within (WR) and between (BR) components[45,46]. We examine these components for 165 countries, during a pre-COVID (2000–2019) period. We assumed there would be pandemic-related short-run impacts on infrastructure availability as retrieved by NTL[47], which will course correct over time[48]. Socioeconomic data availability constraints reduced the number of countries we analyzed for each question. For the first question, we use data for 147 countries, offering an empirical perspective on whether urbanization and economic development are conducive to increased infrastructure inequalities. The second question tests a hypothesis of increasing infrastructure inequalities under the purview of global urbanization and economic development, which contrasts with existing theories on spatial inequalities suggesting an inverted-U or cyclical evolution[49,50]. Our third question tests for the

long-run predictability of infrastructure inequalities. We forecast changes in infrastructure inequalities under five future development scenarios, i.e. shared socio-economic pathways (SSP), for 138 countries to understand the implications of future urbanization and economic development with a focus on nine world regions. To address our final question, we compare our forecasted infrastructure inequalities for 2050 with economic inequality forecasts[51] under different SSPs for 121 countries.

In this work, we show long-run positive associations between infrastructure inequalities and urbanization as well as economic development, and historical increases in infrastructure inequalities across most of the countries examined. Our infrastructure inequality forecasts highlight greater inequality increases in countries from the global south than in the global north, and that the future changes in infrastructure inequalities can differ from forecasted changes in income inequalities.

## Results

### Urbanization-infrastructure inequalities association

On average, countries with higher WR inequality levels had higher BR inequality levels (Fig. 1a). WR inequality is positively correlated with BR inequality ($\rho = 0.71$, $p$-value $< 0.01$). This association suggests that nighttime lights-derived infrastructure tends to concentrate in space, regardless of the geographical scope, i.e., local or regional. This is surprising because within- and between-region economic inequalities often show contrasting trends. Still, some deviations existed from this average relationship, which we attribute partly to the spatial scale at which we compute WR and BR inequalities as well as differences in land area. Pearson's correlation coefficients estimated between WR and BR inequalities across spatial scales (varying lattice grid resolution in degrees) showed a strong positive association only at some intermediate scales such that at lower and higher grid resolutions the strength of positive correlation decreased. These variations, nevertheless, suggest optimal scales to simultaneously examine WR and BR inequality levels, across countries. Similarly, land area can influence infrastructure distributions in a country and simultaneously the level of measured inequality. We found a predominant influence on BR inequality: (log) land area ($km^2$) is more correlated to BR inequality than WR inequality (Supplementary Figs. S1 and S2).

Urbanization and economic development were significant predictors of long-run infrastructure growth. Results showed that these predictors are positively associated with infrastructure inequality (Fig. 1b–d), which was robust at a vast majority of spatial scales. Controlling for (log) land area ($km^2$), these related variables—urban demographic shares (%) and (log) GDP per capita based on purchasing power parity (PPP) at constant 2017 international \$—cumulatively led to a greater increase in BR inequality than WR inequality (Supplementary Table S1). Still, the combined variation in urbanization and economic development explained 64% and 46% of the total variation in WR and BR inequality, respectively (Supplementary Tables S2, 3). In comparison, the land area variable explained 7% and 25% of the total variation in WR and BR inequality, respectively. Uncertainties in NTL, urbanization (owing to varying urban definitions), and economic data can influence and cause deviations from the average relationship. However, our results also suggest variations in the country-level inequality levels such that countries with similar levels of urbanization and economic development can still have varying levels of infrastructure inequalities.

Assuming NTL infrastructure is tantamount to economic output, we can expect inequality levels to show a cyclical evolution characterized by divergence and convergence trajectories, as suggested by existing theories on regional inequalities[49,52]. Our results present no evidence of such non-linear patterns in the infrastructure inequality estimates. Instead, they suggest that infrastructure inequalities are positively linked with urbanization and economic growth.

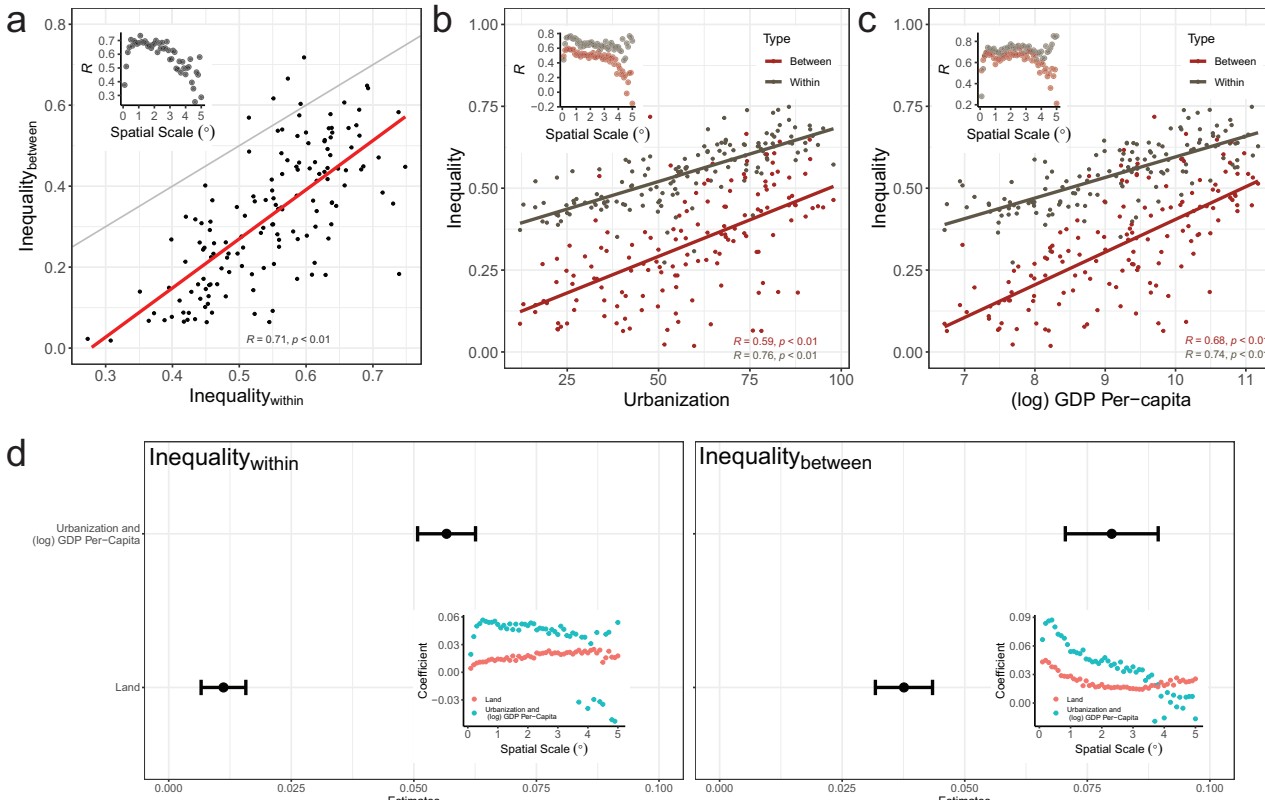

**Fig. 1 | Cross-sectional variations in nighttime lights-derived inequality levels for 147 countries in the year 2015, estimated using a 0.5° lattice grid.** An increase in within-region inequality accompanies an increase in between-region inequality (and vice versa) (**a**). The gray line shows the 1:1 diagonal. Within- and between-region inequalities are correlated with urbanization (urban share of the total population (%)) and (log) GDP per capita based on purchasing power parity (PPP) at constant 2017 international dollars (**b**, **c**). Combined variations in urbanization and (log) GDP per-capita derived from principal component analysis (first principal component) is positively associated with inequality levels (**d**). Error bars show the 95% confidence interval based on bootstrap standard errors (1000 replications). Inset plots in **a–c** show Pearson's correlation coefficients ($R$) obtained from analyzing the respective relationships with lattice grids of varying resolutions $\in [0.1, 0.2, ..., 4.9, 5.0]$, specifying spatial scales (°), and in (**d**) show regression coefficients obtained using inequality estimates at varying spatial scales.

Consequently, contemporary global trends—oriented towards increased urbanization and economic development—set an expectation for increasing infrastructure inequalities over time, on average.

**Dominant increasing infrastructure inequalities trend**

The results of our temporal analysis showed a dominant increasing inequality trend across countries (Fig. 2). BR (WR) infrastructure inequality showed a statistically significant ($p < 0.05$) monotonic trend in 140 (115) countries out of the 165 countries examined (Supplementary Fig. S3). In 130 (119) countries, BR (WR) inequality increased at increasing rates (Fig. 2). Declines in inequality were rare but distinctively evident in Syria and Yemen, where conflicts leading to internal population displacement and economic declines have caused declines in NTL and, consequently, infrastructure inequalities. Our multi-scale analysis of changes in infrastructure inequalities suggests that the observed increase in infrastructure inequalities is robust to varying spatial resolutions for inequality measurement. In sum, these trends and patterns provide further evidence that infrastructure inequalities are integral to urbanization and economic growth and decline under large-scale perturbations such as war and conflicts. However, a detailed examination of the WR and BR inequality time series suggested that stochastic processes govern their evolution dynamics, which can also explain the rare declines in some other countries.

Random walk characterized much of the year-on-year changes in infrastructural inequalities. Hypothesis testing using the augmented Dickey-Fuller test supported this finding. Results showed unit root in changes in inequality for over 90% of the countries examined (Supplementary Fig. S4). Unit root indicates that a deterministic process cannot explain how infrastructure inequality would evolve. It suggests infrastructure inequality dynamics have persistence; inequality in a given year depends on inequality in the preceding year and a random stochastic component. These characteristics impede our ability to predict the short-run evolutionary dynamics of BR and WR inequality reasonably.

By contrast, in the first-difference models, urbanization and economic development explained WR and BR inequality changes between 2000 and 2019 (Supplementary Table S4). Still, we found that urbanization and economic development better explain changes in BR inequality than WR inequality. Controlling for Δ (log) sum of lights, a 1% increase in Δ GDP per-capita was associated with a 0.042 (0.046) unit increase in Δ WR (BR) inequality. Δ Urbanization had a similar association with Δ WR and BR inequality. A unit increase in Δ urbanization was associated with a 0.002 unit increase in Δ WR and BR inequality. These positive associations suggest that urbanization and economic development are associated with infrastructure inequality in the long run.

Panel autoregressive distributive lag model (ARDL) estimations confirmed long-run associations between infrastructure inequalities and urbanization as well as economic development. Panel unit root testing based on Im–Pesaran–Shin (IPS) test[53] suggested that all variables (except urbanization that is $I(0)$) are $I(1)$, i.e., integrated to first order, at the 0.01 level. Pooled mean group (PMG) and dynamic fixed effect (DFE) estimates showed significant and positive long-run

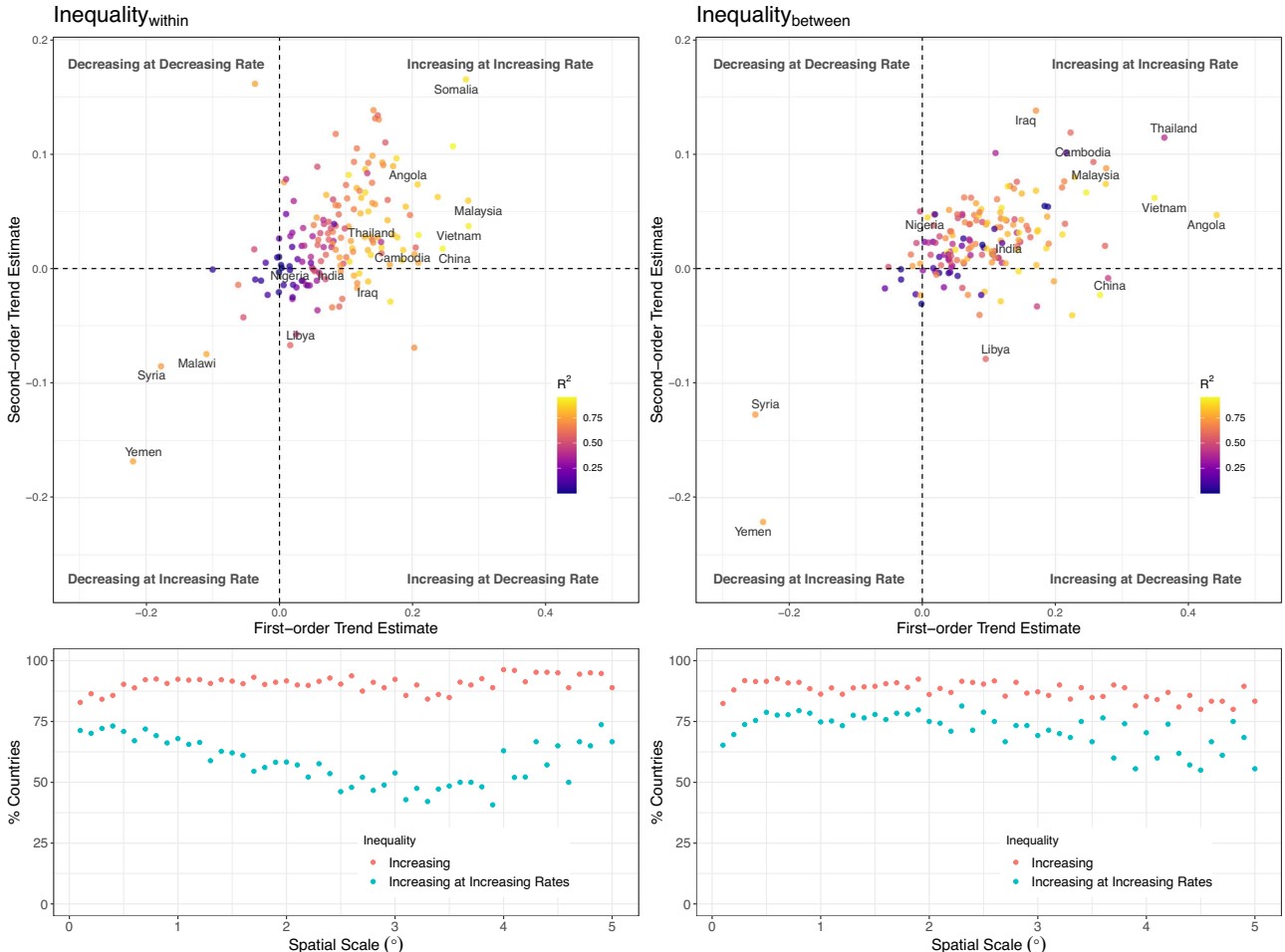

**Fig. 2 | Changes in within-region (Inequality_within) and between-region (Inequality_between) inequality during 2000–2019 across 165 countries, estimated using a 0.5° lattice grid.** Top panel shows first-order and second-order trend estimates from ordinary least squares (OLS) fit for 165 countries. Bottom panel shows the percentage of countries where within-region (left) and between-region (right) inequalities are increasing or increasing at increasing rates.

associations of WR and BR inequalities with urbanization, NTL, and economic development at the 0.01 significance level (Supplementary Tables S5–8). Across all models, $\phi$ estimates were significant and negative at the 0.01 level. Mean group (MG) estimates were inconsistent, providing support to our expectation that the long-run associations are constant across countries. Hausman tests suggested that PMG is more consistent and efficient than MG at the 0.01 level for WR and BR inequality.

In sum, our results emphasize a dominant trend of increasing infrastructure inequalities and rare cases of decreases in inequalities. Rising WR and BR inequality were more prominent in the world's developing regions than in developed regions. Rapid increases in inequalities, i.e., WR and BR inequality increasing at increasing rates, were particularly evident in Southeast Asian countries with substantial urban primacy that are also rapidly urbanizing: Malaysia, Vietnam, Thailand, and Cambodia. Our results suggest that urbanization may intensify urban primacy with increasing infrastructure inequalities. Despite the greater magnitude and pace of urban expansion, we found positive but relatively linear inequality changes in China and India. Additionally, random walk characterized short-run inequality changes, which could have been due to noise in the NTL time series that influences our subsequent inequality measurement. As a result, reasonably predicting short-run inequality changes became difficult. Nevertheless, we found a significant influence of urbanization and economic development on inequality in the long run.

### Greater infrastructure inequality increases in the global south

Consistent with inequalities at the national scale, within (local) and between (regional) inequality levels, estimated using a 0.5° lattice grid, averaged for world regions showed positive associations (Fig. 3a). We found strong positive associations between BR and WR inequalities across time in all world regions ($\rho > 0.7$, $p$-value < 0.01) except for North America ($\rho = 0.58$, $p$-value < 0.01). Furthermore, BR and WR inequality levels were more similar in three regions, i.e., India (IND), China (CHN), and North America (NAM), than in other regions. Of all world regions, CHN showed the most rapid change in the two types of inequalities, which we attribute to rapid urbanization and economic growth based on our regression results. Nevertheless, our forecasts across SSPs showed considerable differences from these historical patterns.

Results of the SSP-based forecasting showed that future WR and BR inequality will evolve differently in different regions but with greater changes in the global south than the global north (Europe and Central Asia (ECA) and NAM) (Fig. 3b, c). Broadly, we group these differences between world regions into two categories: (1) decoupled WR and BR inequality changes, and (2) simultaneous increases in WR and BR inequality. Six out of nine regions showed decoupled changes, i.e., changes in the two inequality types differ in direction (Fig. 3c). Of these six regions, regions in the global north showed minimal changes.

North American countries, i.e., the United States and Canada, had existing high levels of infrastructure inequality, and these levels are

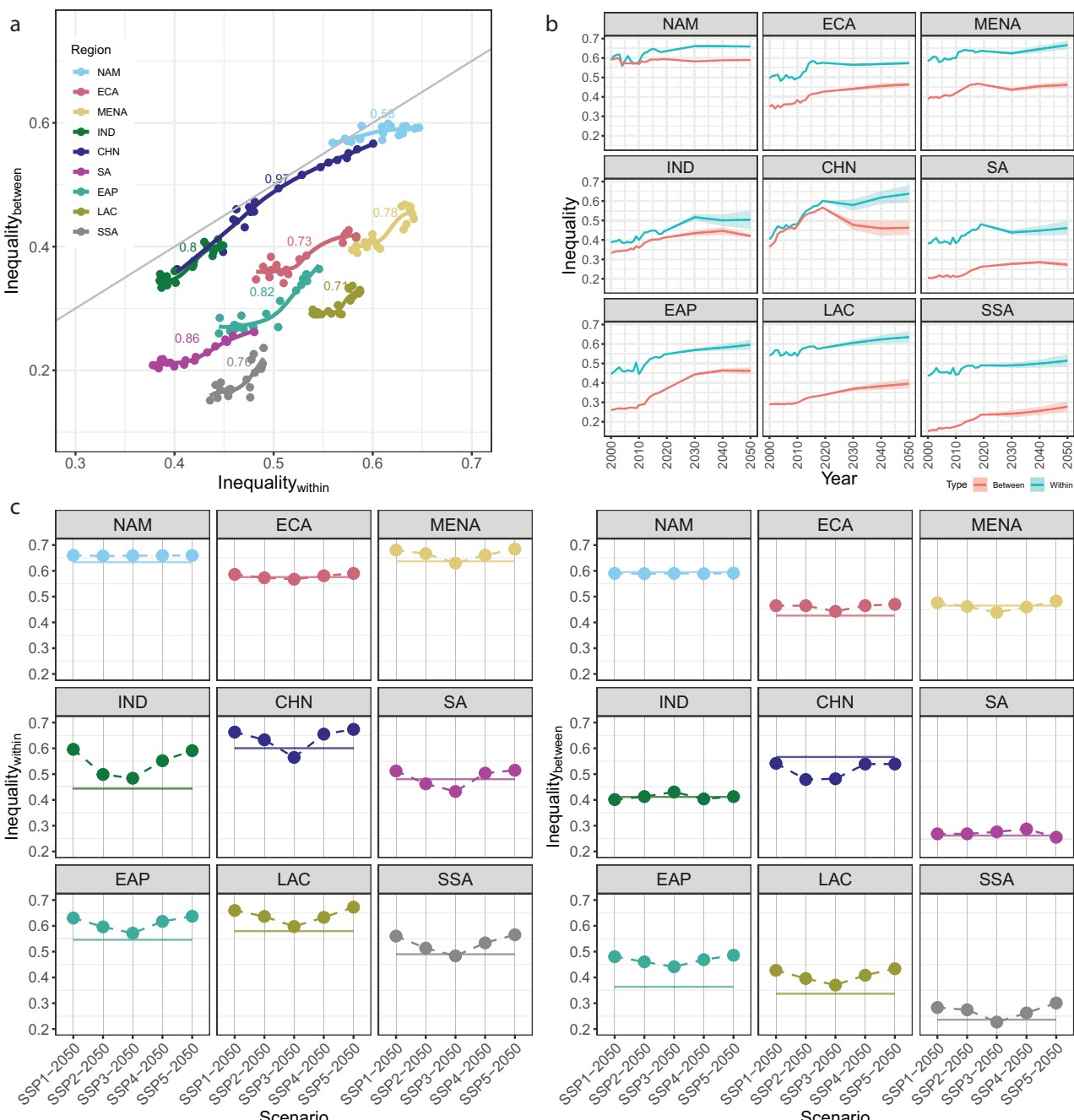

**Fig. 3 | Historical and forecasted infrastructure inequalities aggregated across nine world regions.** Within- (Inequality$_{within}$) and between-region (Inequality$_{between}$) inequality, estimated using a 0.5° lattice grid, from 2000 to 2019 aggregated across nine world regions by using estimates for 138 countries (**a**): sub-Saharan Africa (SSA), Latin America and the Caribbean (LAC), South Asia excluding India (SA), East Asia and Pacific excluding China (EAP), Middle East and North Africa (MENA), Europe and Central Asia (ECA), North America (NAM), India (IND), and China (CHN). Lines show LOESS regression fits, and numeric labels show Spearman's correlation coefficients (*p*-value < 0.01). Inequality forecasts averaged across world regions using estimates for 138 countries, highlighting substantial increases in world regions with existing urban primacy problem (**b**). Lines beyond 2019 show forecasts under business-as-usual shared socio-economic pathway (SSP), i.e., SSP2. Line intervals show the forecasted range across SSP scenarios. Changes in inequality will be least in SSP3 and greatest in SSP5 (**c**). The figure shows changes in inequality under five SSPs from 2019 to 2050. Solid lines depict 2019 levels.

likely to persist with a marginal increase in WR inequality by 2050. Similarly, future changes in inequality were marginal in the ECA region. In contrast, India showed substantial increases in WR inequality with little or no change in BR inequality. The decoupled future change in India is unexpected as its current urban population of 483 million is expected to increase ~1.8 times between now and 2050. Based on our panel ARDL estimations, we expect concomitant increases in BR and WR inequality levels with the impending urbanization. We found a similar unexpected pattern in China, but with decreasing BR inequality. These results can be attributed to country-specific characteristics, which we account for in the forecasting model.

We found simultaneous future increases in WR and BR inequality in world regions known for countries with substantial urban primacy (Fig. 3b, c). Greater urban primacy implies a greater development imbalance between urban areas and a pronounced core-periphery urban structure. Our forecasts suggest that this phenomenon can become more

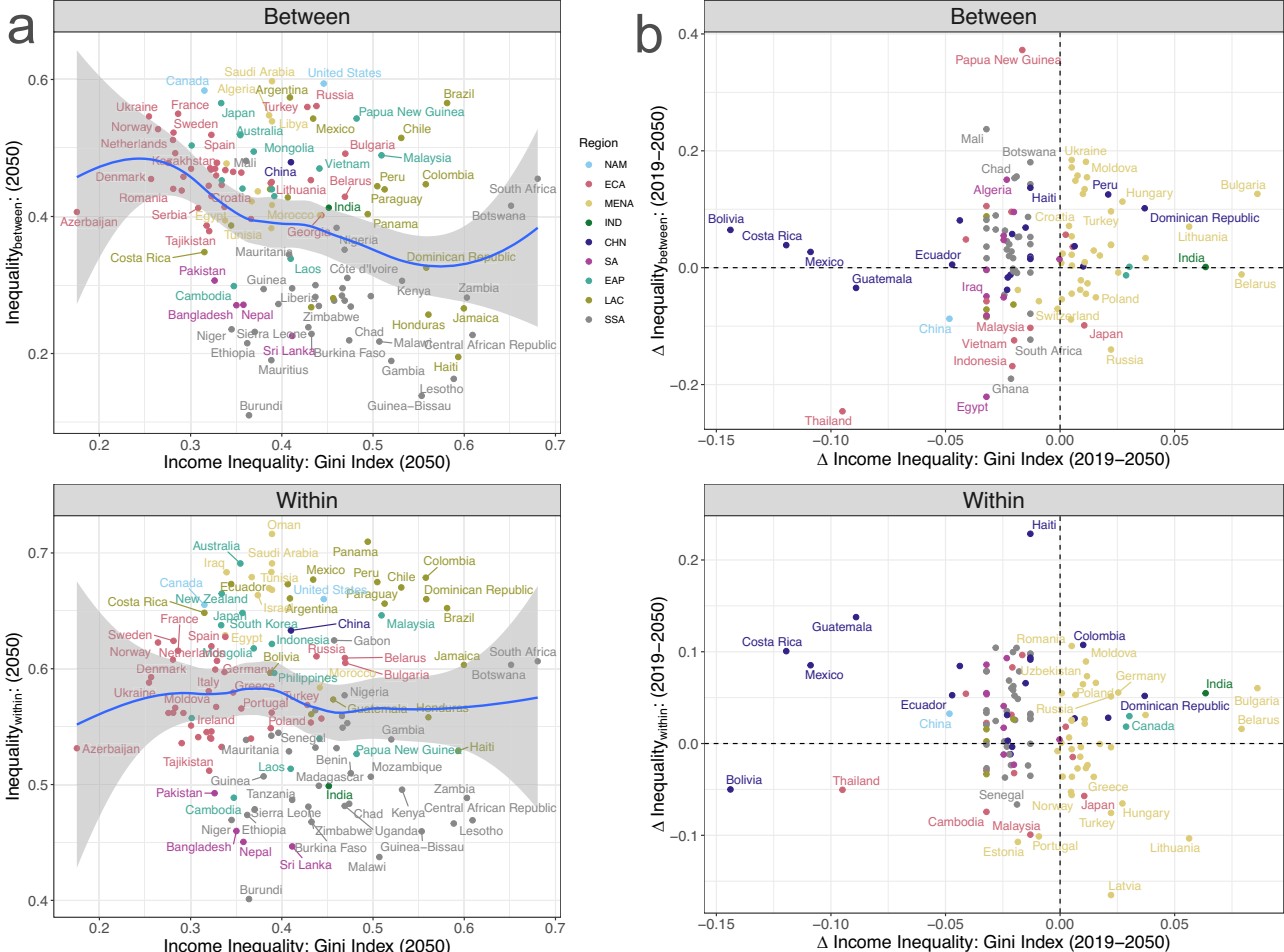

**Fig. 4 | Comparing forecasted between- and within-region inequality with forecasted income inequality.** Comparing forecasted 2050 between (Inequality_between) and within (Inequality_within) region inequality levels and income inequality levels (**a**) and associated changes from 2019 to 2050 (**b**) under the business-as-usual socioeconomic development scenario (SSP2) for 121 countries classified into nine regions (**a**): sub-Saharan Africa (SSA), Latin America and the Caribbean (LAC), South Asia excluding India (SA), East Asia and Pacific excluding China (EAP), Middle East and North Africa (MENA), Europe and Central Asia (ECA), North America (NAM), India (IND), and China (CHN). Blue line in **a** represents locally estimated scatterplot smoothing (LOESS) fit and the shaded region in gray represents 95% confidence interval.

pronounced in Latin America (LAC), East Asia (EAP), and Sub-Saharan Africa (SSA) regions, with increasing future increases in inequality. Nevertheless, inequality will rise within and between urban areas.

Future changes in inequality will also depend upon the socio-economic development pathways that countries would take and more so in the global south (Fig. 3c and Supplementary Table S9). Our results showed a clear difference in future inequality outcomes vis-à-vis SSP scenarios in all regions except for the more-developed NAM and ECA regions, where WR and BR inequality outcomes remain similar across scenarios (Fig. 3c). Future changes in inequality will be minimal under the SSP3 scenario, owing to low urban and economic growth and low infrastructure inequality levels. The SSP narrative emphasizes reduced trade dependencies with lower human and technological investments under the SSP3 scenario[54,55]. Through their impact on urban and economic growth, these characteristics lead to minimal changes in infrastructure inequality. In contrast, we found the greatest rise in inequality across all regions under the development and trade-intensive SSP5 scenario. Altogether, our results point to an indirect positive link between international trade (and investments) and infrastructure inequalities. Of the five scenarios, SSP4 is the most relevant scenario in terms of the urbanization-inequality association implications. Based on the SSP narrative, we expect considerable disparities in human development outcomes, economic opportunities,

and power dynamics under this future scenario[51]. Forecasts suggest greater increases in infrastructure inequality levels in EAP, IND, and LAC than in other regions (Fig. 3c and Supplementary Table S9)

## Future infrastructure and income inequalities

Our results highlight that urban infrastructure inequalities can differ from economic inequalities in terms of characteristics and dynamics. Forecasts across 121 countries showed heterogeneity in the levels and changes in infrastructure inequalities between countries with similar income inequality levels (Fig. 4a) and changes (Fig. 4b). This observed heterogeneity was consistent across all future scenarios (Supplementary Figs. S5, 6). Moreover, it highlighted the absence of any strong positive association between infrastructure inequalities and income inequalities. There can be at least four reasons for the absence of an expected positive correlation. First, the measured and forecasted infrastructure inequalities are geographical (between places and regions), whereas income inequalities are between individuals/households, and these scale differences can cause the observed differences. Second, income inequality data tends to have quality issues, which can bias the correlation. Third, assuming that infrastructure inequalities influence income inequalities, there still can be large time lags to the extent that the influences are realized at inter-generational scales.

Fourth, infrastructure inequalities have different characteristics and dynamics than economic inequalities. Regardless of the reasons, our results suggest that infrastructure inequalities require a different approach and analytical lens than economic inequalities.

## Discussion

Our study uses satellite data to provide robust evidence of increasing urban infrastructure inequalities. We show a dominant trend of rising infrastructure inequalities in many countries. This trend contradicts economic geography and regional science theories that suggest a cyclical evolution of economic inequalities. The cyclical evolution has many explanations but is generally linked to economic expansion and contraction phases. Regional inequality rises in the expansion phase and diminishes in the contraction phase. Our results suggest that this is not the case with infrastructure inequality. One explanation for this inconsistency is that infrastructure changes are generally unidirectional: infrastructure, once built, is rarely removed. As a result, infrastructure inequalities rise during the growth phase and stay constant otherwise. Our results support this explanation by revealing a dominant trend of rising inequalities both between and within regions. Moreover, they set an expectation that infrastructure inequalities may be associated with regional (economic) inequalities during the divergence phase, i.e., when regional economic inequalities rise. Conversely, results show that inequalities decline with large-scale perturbations—such as war and conflict—that affect infrastructure availability.

Further, our findings show that urbanization and economic development underpin rising infrastructure inequalities in the long run. Population growth in urban areas and economic development are well-known drivers of urban land and infrastructure expansion[56,57]. We find that these drivers also increase infrastructure inequalities within and between regions such that more urbanized and economically developed countries tend to have greater infrastructure inequalities. Consequently, rapidly urbanizing countries tend to show a faster increase in infrastructure inequalities. For example, we find that inequalities rapidly increased in China and some Southeast Asian countries during the 2000–2019 period. Increasing inequality can be a problem for urbanizing countries, as infrastructure inequalities due to their durable nature can perpetuate or amplify socioeconomic inequalities[25,58]. This is especially concerning for lower-income countries that are either in the early stages of urbanization or have yet to urbanize[59,60]. At the same time, these countries have a unique opportunity to mitigate these risks from the outset. Infrastructure development has been frequently argued to be an essential driver of global sustainable development[7,61]. However, our findings challenge this view by highlighting the spatial inequalities it can entail. Instead, they stress the need for intentional infrastructure development prioritizing spatial equity in all countries regardless of their stage of urbanization or economic development, rather than focusing solely on aggregate development and assuming inequalities can be addressed later through equity measures[4,41].

Moreover, the growing infrastructure inequalities within and between regions underscore the importance of addressing these disparities in the context of urbanization with a focus that extends beyond urban areas. This contrasts with the normative emphasis, such as in SDG 11, on promoting inclusive development within urban areas to advance sustainable development[62]. Beyond these policy implications, our results suggest a potential tradeoff between spatial efficiency gains from urbanization and spatial equity gains. To some extent, this tradeoff may limit the effectiveness of national urbanization policies aimed at reducing spatial inequalities[63].

Our forecasts suggest that infrastructure inequalities will predominantly increase in the Global South. Under the business-as-usual scenario (SSP2), future urbanization and economic development will exacerbate inequalities in East Asia, Latin America, Sub-Saharan Africa,

and South Asia with greater extents compared to other regions. These increases may also intensify the existing urban primacy problem, whereby resources are inequitably concentrated in primate urban areas, widening disparities in quality of life[64]. This is particularly concerning given recent evidence that suggests a decoupling between urbanization and development in many rapidly urbanizing countries in the Global South[60,65]. Even if countries in these regions urbanize without corresponding economic development, our regression results and inequality forecasts indicate that infrastructure inequalities will still rise. This could lead to a problematic outcome where low levels of development coincide with high levels of inequality, increasing vulnerability to stressors such as climate change and social instability[66], influencing climate adaptation actions[67]. Additionally, if countries follow growth trajectories consistent with the SSP4 narrative, our forecasts suggest a substantial rise in infrastructure inequalities. Under this scenario, infrastructure inequalities within and between regions may intersect and further exacerbate inequalities in human development outcomes.

Our findings have important implications for our understanding of urbanization and its socio-environmental implications. Urbanization is primarily viewed as a demographic process in which the urban population share rises over time, accompanied by increased average economic activity. As urbanization progresses, the number and physical size of urban areas expand, and infrastructure stock rises within and beyond urban areas. While urbanization is often seen as a driver of infrastructure growth, our results suggest that this process also contributes to rising infrastructure inequalities. In this context, we argue that infrastructure inequalities are fundamentally intertwined with urbanization and are inherently urban in nature. Moreover, our forecasts indicate that future scenario development and analysis focused on socio-economic development and climate change impacts may overlook other types of urbanization-related inequalities. While SSPs remain the dominant and widely-used global future scenarios, we need additional research to understand the causes and implications of infrastructure inequalities. To address this, multi-scale, multi-context and geographically diverse understanding of inequalities built into the urbanization process will be essential.

Finally, our study highlights three key additional areas for future research. First, although this study accounts for population migration through its effects on geographic shifts in population distribution, the nuances of migration and its relationship with infrastructure inequalities[68] require further exploration. Second, future analyses of infrastructure inequalities vis-à-vis urbanization can be advanced by capturing multiple dimensions of infrastructure availability, accessibility, and quality, and assessing their spatial and social distributions. This could be achieved in part by integrating satellite remote sensing data with ground-based surveys or infrastructure service datasets. It remains unclear whether the patterns and dynamics of aggregate infrastructure inequalities highlighted in this study are consistent across other aspects of infrastructure. Third, future research could explore how using administrative boundaries at different administrative scales as well as computational methods delineating regions might influence infrastructure inequality estimation.

## Methods

### Data

We used two sources of NTL: (1) The Defense Meteorological Program (DMSP)/Operational Line-Scan System (OLS) and (2) Visible Infrared Imaging Radiometer Suite (VIIRS) Day-Night Band (DNB). Annual stable lights product from DMSP/OLS extends from 1992 to 2013 but has temporal inconsistencies[69]. We used a global harmonized NTL dataset (spatial resolution of ~1 km at the equator) for years 2000 to 2013[70]. Given the superiority of VIIRS DNB over DMSP/OLS[71], we used VIIRS DNB monthly average radiance (spatial resolution of ~0.5 km at the equator) time series from 2012 to 2019[72]. We reprocessed VIIRS DNB at

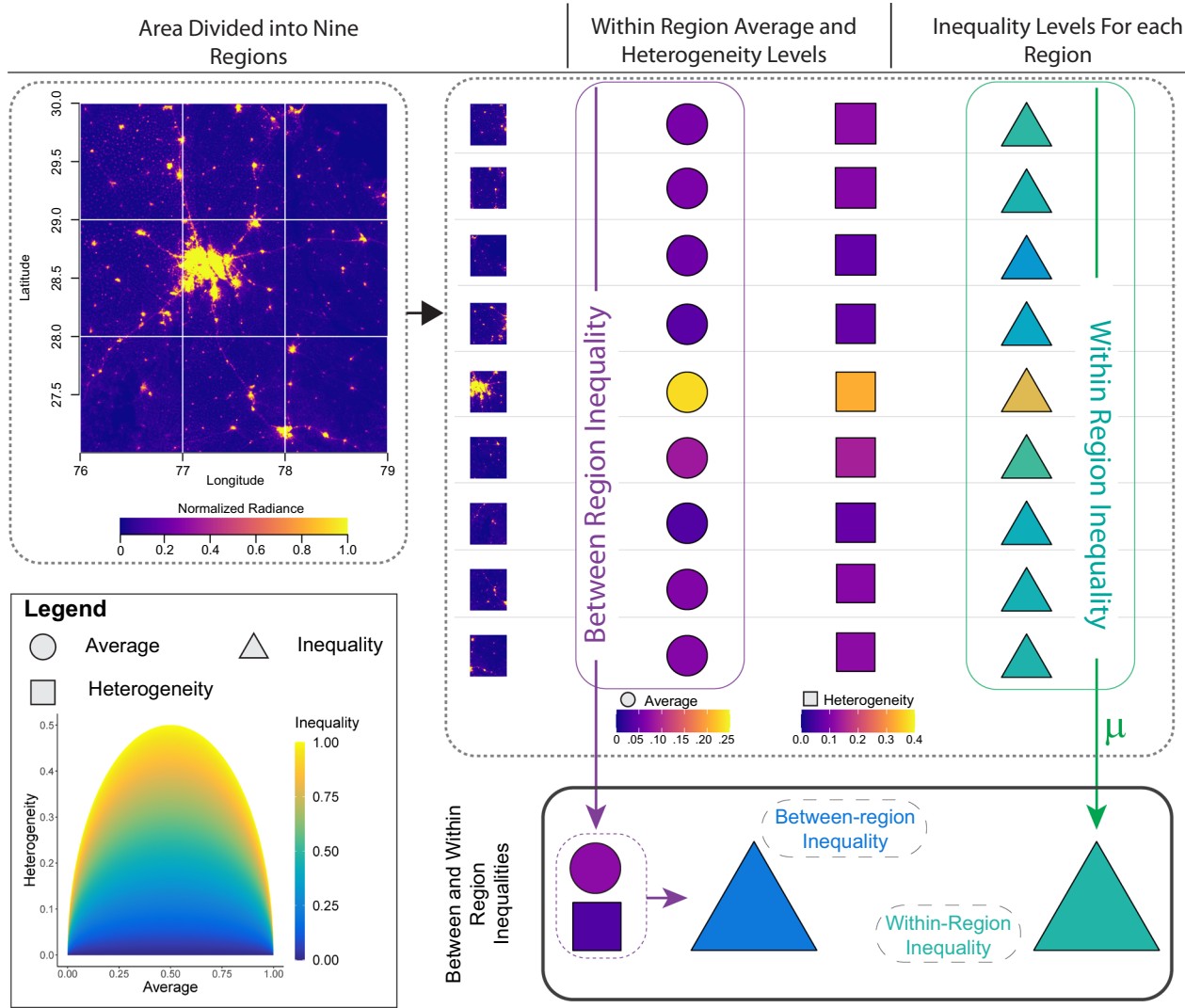

**Fig. 5 | Illustrating between- and within-region inequality measurement using nighttime lights radiance truncated with an upper-bound.** The entire area is divided into regions and average (circles) radiance, associated heterogeneity (squares) and inequality levels (triangles) are estimated for each region. Between-region inequality is estimated using average radiances across regions and within-region inequality is estimated as the average of inequality levels within each region.

the monthly frequency to calculate average stable radiance, an indicator of infrastructure availability, at an annual frequency[73]. Our reprocessing removed outliers in the per-pixel NTL time series using an outlier removal algorithm[72], substituted average radiance values with stray-light corrected radiance values where the average monthly radiance dataset recorded zero values, and applied a Seasonal and Trend decomposition using Loess (STL) to discard seasonality and noise in the time series[74]. Previous research attributed seasonality in VIIRS NTL to biophysical factors irrelevant to our measurement of infrastructure inequality[75]. Our reprocessing of the VIIRS time-series images removed seasonal fluctuations and yielded per-pixel non-linear trends. We used the monthly outputs to calculate annual VIIRS NTL images from 2012 to 2019. We addressed DMSP/OLS and VIIRS data-related inconsistencies, owing to differences in spatial and radiometric resolutions, while measuring inequality.

A global population distribution raster dataset at ~1 km spatial resolution is available from Landscan, which provides ambient population distributions (daily average) annually[76]. We used population counts to calculate a weighted measure of infrastructure inequality. This approach obviates examining per-pixel changes in population, which can be unreliable. Instead, it ensures that infrastructure inequalities account for population distributions.

We evaluated urbanization and GDP per capita (based on purchasing power parity (PPP) at constant 2017 international $) as the two principal predictors of infrastructure inequality. Socioeconomic data has better global and temporal coverage at the national scale than at the sub-national scale. We accessed national statistics from the open data repository at the World Bank using the *WDI* package in R[77].

We used the SSP database (https://secure.iiasa.ac.at/web-apps/ene/SspDb/) to gather future urbanization, economic growth, and economic inequality projections under different socio-economic development pathways. GDP per-capita time series in the SSP database uses a 2005 base. We rebased GDP per-capita forecasts under each SSP scenario to match with the 2017 base year in historic GDP per-capita time series from the World Bank for each country. Rebasing involved normalizing GDP per-capita forecasts by levels in 2010 and multiplying the normalized values by 2010 GDP per-capita values from the World Bank series.

### Inequality measures
We advanced and applied a heterogeneity-conditional-on-mean measure[39] to estimate BR and WR inequality across countries and over time (Fig. 5). This measure leverages first- ($\mu_r$) and second-order

($\sigma_r$) moments of NTL distributions, and requires pre-specified regions. Region demarcation can be based on administrative divisions or spatial discretization methods[78]. However, differences in region demarcation across countries per se can become a driver of measured inequality. In the case of spatial discretization methods, the identified regions are likely to emphasize between-region inequality over within-region inequality[78], complicating comparisons over time and across countries. Our inequality estimation used statistical moments of NTL distributions calculated with a uniform 0.5° lattice grid that spans globally but excludes grid cells not covering land area. Still, for robustness analysis of the findings, we used lattice grids of varying resolutions $\in [0.1, 0.2, \ldots, 4.9, 5.0]$, specifying spatial scales (°) and estimated inequality across spatial scales. Note that inequality estimation with coarse-scale lattice grids may not work for smaller-sized countries (Supplementary Fig. S7). Similarly, using fine-scale grids, closer in spatial resolution to NTLs resolution may introduce a bias towards either within or between-region inequality, complicating analysis across countries. Hence, we expected consistency of our results at some optimal spatial scales where the select number of countries are globally representative. For a grid cell (region $r$), we evaluated NTL geographic concentration level ($I_r$) using Eq. (1) that represents the ratio of observed heterogeneity to the corresponding theoretical maximum heterogeneity.

$$I_r = \frac{\sigma_r}{\sqrt{\mu_r \times (1 - \mu_r)}} \tag{1}$$

Where all the moments were normalized by NTL upper bound radiance thresholds ($T$), which is intended to reflect diminished utility of infrastructure availability beyond a certain threshold. Still, we have limited knowledge of the functional form underlying the lights-utility relationship. Therefore, we assumed data-specific thresholds to be 63 for DMSP/OLS NTL, the highest value recorded in the dataset, and 25 nWsr$^{-1}$cm$^{-2}$ for deseasonalized VIIRS NTL. Estimated inequality levels are sensitive to these thresholds[39] and hence we kept them constant across all countries, for consistency. Additionally, our measurement assumes existence of infrastructure deficits and focus on the preferential allocation within and across regions given the deficits. We calculated WR inequality ($Inequality_{within}$) as a population ($p_r$) weighted average measure of $I_r$ across regions using Eq. (2).

$$Inequality_{within} = \frac{\sum_{r=1}^{n} p_r I_r}{\sum_{r=1}^{n} p_r} \tag{2}$$

The $\mu_r$ distribution in a given country suggests inequality between regions. We quantified BR inequality ($Inequality_{between}$) using a population-weighted first- ($\mu_{\mu_r}$) and second-order ($\sigma_{\mu_r}$) moments of $\mu_r$ in Eq. (3). We calculated WR and BR inequality for 165 countries and territories listed in the global administrative areas (GADM) dataset version 3.6[79] and across time.

$$Inequality_{between} = \frac{\sigma_{\mu_r}}{\sqrt{\mu_{\mu_r} \times (1 - \mu_{\mu_r})}} \tag{3}$$

We combined DMSP/OLS-based and VIIRS-based inequality time series to construct a twenty-year time-series record, addressing the inconsistencies. The average difference in inequality levels for 2012 and 2013, the overlapping years between the two datasets, was used to adjust the VIIRS-derived inequality levels.

## Cross-sectional analysis

Cross-sectional distributions can suggest long-run relationships between inequality and corresponding predictors. We examined urbanization and economic growth as the key predictors of infrastructure inequalities and quantified the relationships using ordinary least squares (OLS) regression and bootstrapped regression estimates as a non-parametric approach to uncertainty estimation. We then evaluated whether there is an inverted U-shaped relationship between inequalities and the two potential predictors. The cross-sectional analysis focused on the year 2015 and a sample of 147 countries (selected based on data availability).

## Spatio-temporal analysis

We examined country-specific trends in WR and BR inequality using the Mann–Kendall trend test and characterized the trends based on first- and second-order parameters of a quadratic fit. Here we examined 165 countries specified in the GADM dataset. We used the augmented Dickey-Fuller test to test for a random walk. Next, we estimated first-difference models to examine the long-run relationships between inequalities and the two potential predictors by focusing on changes between 2000 and 2019 (Eqs. (4) and (5)). Here, limited data availability led to a sample of 144 countries.

$$\Delta I_i = \Delta Urbanization_i + \Delta GDP_{pc_i} + \varepsilon_i \tag{4}$$

$$\Delta I_i = \Delta Lights_{sum} + \Delta Urbanization_i + \Delta GDP_{pc_i} + \varepsilon_i \tag{5}$$

Finally, we estimated panel ARDL models to examine short-run and long-run associations using a balanced panel of 143 countries[80,81]. Equation (6) specifies a panel ARDL.

$$I_{it} = \sum_{j=1}^{p} \lambda_{ij} I_{i,t-j} + \sum_{j=0}^{q} \delta'_{ij} X_{i,t-j} + \mu_i + \varepsilon_{it} \tag{6}$$

where $i$ and $t$ are country and year, respectively. $p$ and $q$ are the time lag length on Inequality and the explanatory variable ($X$), respectively. $\mu_i$ are the country-specific effects. When variables are cointegrated and $\varepsilon_{it}$ are zeroth-order integrated errors ($I(0)$), i.e., errors do not accumulate past errors, Eq. (7) is a reparameterization of Eq. (6). This specification implies an error correction model (ECM) where the variables' short-run dynamics are deviations from the long-run equilibrium relationship.

$$\Delta I_{it} = \phi_i (I_{i,t-1} - \theta'_i X_{it}) + \sum_{j=1}^{p-1} \lambda^*_{ij} \Delta I_{i,t-1} + \sum_{j=0}^{q-1} \delta'_{ij} \Delta X_{i,t-j} + \mu_i + \varepsilon_{it} \tag{7}$$

$\phi_i$ is the error-correcting speed of the $(I_{i,t-1} - \theta'_i X_{it})$ term and suggests a long-run relationship when $\phi_i \neq 0$. $\theta'$ specifies the long-run relationships. We estimated and compared the mean-group (MG), pooled mean group (PMG), and dynamic fixed effect (DFE) estimations[82]. Because urbanization and log GDP were highly correlated in the panel ($\rho = 0.8$; $p$-value < 0.01), we estimated Eq. (6) for these variables individually.

## Forecasting inequality

Building upon findings from the cross-sectional and temporal analysis, we developed a random forest regression-based forecasting model[83]. Random forests are an ensemble learning method that yields high accuracy predictions. We trained the machine learning algorithm on a pooled dataset with four regressors: urbanization, GDP per-capita, land area, and world region. The pooled dataset contained values from 2000 to 2015. We assessed model accuracy with the coefficient of determination ($R^2$) estimated between predicted and observed inequalities using an out-of-sample forecast for the subsequent years (2016–2019). $R^2$ values exceeded 0.90 and 0.80 for BR and WR inequalities, respectively. Using future urbanization and GDP per-capita levels under the five SSP scenarios, we forecasted WR and BR

inequality for 138 countries. Lastly, we calculated and interpreted average inequalities across nine world regions: sub-Saharan Africa, Latin America and the Caribbean, South Asia excluding India, East Asia and Pacific excluding China, Middle East and North Africa, Europe and Central Asia, North America, India, and China. Here we compared the forecasted infrastructure inequalities with economic inequalities, from the SSP database[51], for 121 countries.

## Data availability

All the input data used in the analysis are available from sources referenced in the main text. Global harmonized nighttime lights dataset is available from https://doi.org/10.6084/m9.figshare.9828827.v2. Gridded population data is available from https://landscan.ornl.gov/. Global administrative areas dataset is available from https://gadm.org/. Shared socioeconomic pathways (SSPs) dataset including urbanization GDP per-capita, and income inequality forecasts are available from https://tntcat.iiasa.ac.at/SspDb/dsd?Action=htmlpage&page=welcome. Infrastructure inequality estimates across spatial scales generated in this study are available in Zenodo under accession code [https://doi.org/10.5281/zenodo.14303282][84]. Reprocessed VIIRS-derived annual average stable radiance generated in this study has been deposited in Zenodo under accession code [https://doi.org/10.5281/zenodo.14302964][73].

## Code availability

Python, R, and Stata code used in the analysis is available in Zenodo [https://doi.org/10.5281/zenodo.14303282].

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

## Acknowledgements

This study was part of doctoral dissertation research conducted by Bhartendu Pandey at the Yale School of the Environment, also supported by the Yale Institute for Biospheric Studies, and subsequently advanced at Oak Ridge National Laboratory. This manuscript has been authored in part by UT-Battelle, LLC, under contract DE-AC05-00OR22725 with the US Department of Energy (DOE). Research (multi-scale analysis) sponsored in part by the Laboratory Directed Research and Development Program of Oak Ridge National Laboratory, managed by UT-Battelle, LLC, for the U. S. Department of Energy. This work was also supported in part by the U.S. Department of Energy through the Los Alamos National Laboratory. Los Alamos National Laboratory is operated by Triad National Security, LLC, for the National Nuclear Security Administration of U.S. Department of Energy (Contract No. 89233218CNA000001). We thank Esther Parish from Oak Ridge National Laboratory for her feedback on a previous version of this manuscript.

## Author contributions

B.P. conceptualized and designed the research, in consultation with C.B. and K.C.S. B.P. set up the experimental design, compiled and processed the relevant datasets, and led the modeling design and development. All authors interpreted the results. B.P. derived the key findings outlined in the manuscript with feedback from C.B. and K.C.S. B.P. led writing the first draft of the manuscript, which all authors subsequently edited.

## Competing interests

The authors declare no competing interests.
