## [Transparent Peer Review file · Nature Communications]

Rising infrastructure inequalities accompany urbanization and economic development

Corresponding Author: Dr Bhartendu Pandey

Version 0:

Reviewer comments:

Reviewer #1

(Remarks to the Author)

First, let me congratulate the authors on their research effort - this is a timely article. The manuscript examines infrastructure inequalities in light of urbanisation and economic develop using satellite remote sensing data in the form of NTL observations. This is an interesting endeavour, and an original contribution to the extent that it aims to address its main objective through four relevant research questions on infrastructure inequalities as a whole rather than some of its components. This is a well-written paper with a clear structure and sound internal consistency between theory and empirics. Therefore my review focuses on suggestions to improve it conceptually. I think the manuscript will be of immediate interest to people in the field of urban studies, urban sustainability, planning practitioners, and to those working on urban policy. In its present form, I would recommend authors to revise a few shortcomings before this manuscript can be accepted for publication. These are summarised below:

- First, in practice the paper deploys a narrow overview of sustainable development in the context of the relevance of the findings of your study. Here, I think your introduction would greatly benefit from a brief discussion on SDG11. The urban SDG is heavily based on indicators and urban data but this is not without critique. Perhaps you could engage theorists who have flag concerns with such approach. More specifically, there are arguments about the vagueness of the term “urban” as a definition that do not readily ‘fit’ into the urban focus of today’s global policy agendas (see Caprotti et al. 2017). Your article only briefly touches on this (lines 44-47). I encourage you to perhaps offer your definition of urban and what places were effectively considered urban before they were included in your analysis? Furthermore, since sustainability is holistically defined as development that advances human well-being equitably, a more thorough conceptual discussion of inequality would be warranted.
- One limitation of analytical approach is the predictors of long-run infrastructure growth. As discussed by the authors, longer-run effects include crucial mechanisms that would affect infrastructure such as urbanisation rate and economic development, and quite possibly to larger extents. However, population dynamics seem to have been underrepresented, principally migration as a driver of population change. A typical example is internal migration, which in some rapidly urbanising cities in the Global South like Chattogram (formerly known as Chittagong), but many more examples in South and Southeast Asia and Latin America, have experienced incredibly fast population growth mainly through net origin in-migration (see Mia et al 2015; Siddiqui et al 2021). How would one reconcile the long-term shifts in population growth with infrastructure projections? I am unsure as to whether this reconciliation, which directly impacts sustainable development - chief among which, the SDG 11 – but also inequalities within countries, can be incorporated in your forecast exercise. But I think Nature Comms readership would be interested to reading your thoughts.
- Your conclusion touches on an important aspect of urbanisation which was largely silent throughout the rest of the manuscript: that of socio-environmental implications. This point goes back to my first point on sustainable development. If the authors think that infrastructure inequality is somewhat a product of the interplay of demographic, economic, social, and environmental systems, then, in a way, only a simultaneous improvement on these four dimensions can move urbanisation towards sustainable outcomes. This intuitively makes sense for poorer countries, but in rich countries, is an increase in aggregate economic performance (a rise in GDPc) necessary to become more sustainable, and what effects increased sustainability bring to infrastructure inequality? I would argue that rich societies could become more sustainable if they managed to decrease their environmental burdens and existing social inequalities, which would then lead to more prosperous urbanisation. But this in turn raises another question: will efficiency gains in terms of improved sustainable outcomes for urbanisation globally be absorbed by capital owners and thereby contribute to enduring inequality, thus reinforcing civic infrastructural issues (see Sampson 2017).

Sincerely,
Your reviewer

References:

Caprotti, F., Cowley, R., Datta, A., Broto, V.C., Gao, E., Georgeson, L., Herrick, C., Odendaal, N. and Joss, S., 2017. The New Urban Agenda: key opportunities and challenges for policy and practice. *Urban research & practice*, 10(3), pp.367-378.
Mia, M.A., Nasrin, S., Zhang, M. and Rasiah, R., 2015. Chittagong, Bangladesh. *Cities*, 48, pp.31-41.
Sampson, R.J., 2017. Urban sustainability in an age of enduring inequalities: Advancing theory and econometrics for the 21st-century city. *Proceedings of the National Academy of Sciences*, 114(34), pp.8957-8962.
Siddiqui, T., Szaboova, L., Adger, W.N., Safra de Campos, R., Bhuiyan, M.R.A. and Billah, T., 2021. Policy opportunities and constraints for addressing urban precarity of migrant populations. *Global Policy*, 12, pp.91-105.

Reviewer #2

(Remarks to the Author)

This paper studies changing infrastructure inequalities from 2000 to 2019 using remote sensing. It finds the rise of infrastructure inequality in most of the countries, with more significant rise in the global south, especially the ones with more urban primacy. The paper also finds the positive associations between infrastructure inequalities and urbanization and economic development inequalities. The findings are interesting and confirm what we expect. So, what is new? Not much. What do positive associations between infrastructure inequalities and urbanization and economic development inequalities? Who causes whom? Are they bad? The paper raises the research gap that "our understanding of infrastructure inequalities accompanying urbanization is limited." So, what is the contribution of this paper to the knowledge? The paper has not spelled it out, and I therefore am not sure about the contributions and implications. I was also trying to know the research progress in the field, but quite limited.

Reviewer #3

(Remarks to the Author)

Comments:

This study investigates the global rising infrastructure inequalities together with urbanisation and economic development using remote sensing data and future socioeconomic scenarios. From my perspective, the objective of this study is novel and ingenious. When we conducted studies about sustainable infrastructure development, it seemed that we made things complex as we spent too much time and effort on high-accuracy data collection of the infrastructure itself. This study provides a relatively simple but effective approach to assessing sustainable infrastructure, including developing relatively simple models to characterise inequality using NTL to quantify infrastructure, and predicting the future using socioeconomic scenarios. The structure and findings of the study are very clear. My suggestions for this study are more about the spatial data analysis methods and the results' explanations. My suggestion for the current manuscript is Major Revision.

1. Line 53: The study addresses the listed four questions, but in the Abstract, the authors mentioned three findings from the study. Can you please consider making the questions and findings of the study consistent?

2. Line 65: A few explanations should be added here to demonstrate that Earth observation, including remote sensing, has been an essential technology to monitor, assess, and manage sustainable infrastructure development in previous studies. Earth observation for sustainable infrastructure critically contributes to implementing Earth data and infrastructure sustainability.

3. Line 139 and Fig 1: The study area is firstly divided into multiple regions and the regional differences are compared to calculate the inequality. However, the approach of dividing regions has critical impacts on the quantification of inequality, which has been proved in a series of studies about spatial data discretisation and spatial stratified heterogeneity, including doi: 10.1080/15481603.2020.1760434 and many relevant studies. Methods of spatial discretisation generally include many types of supervised and unsupervised approaches. The common approach of spatial discretisation is to determine the optimal combinations of numbers and methods of discretisation that can derive the maximum spatial heterogeneity (OPGD), i.e., inequality in this study. Optionally, authors may also consider a few advanced models for spatial discretisation, such as GOZH and RGD. This means that the inequality calculated in the study will significantly change if you use different methods to divide the study area. I highly recommend authors select a relatively reasonable approach, at least the typical approach of spatial discretisation, to divide the study area for computing inequalities. I believe if you choose other spatial discretisation methods, the study's results will be changed.

4. Line 318: The study investigates infrastructure in urban regions. How did you define urban regions? Which data set of urban boundaries do you use to define urban areas? In the figures, I assume that the authors include both urban and the surrounding regions, which may be rural or non-residential areas, in computing inequality. The selection of rural and non-residential regions may critically impact quantifying inequality.

5. Line 450: The authors mentioned that population is closely linked with urbanisation. This study analyses infrastructure inequality based on spatial heterogeneity and from a spatial perspective. Can you please explain the inequality of the population's share of infrastructure, i.e., infrastructure per person?

6. Fig 5: Can you please consider adding a figure to the statistical summary of findings from Fig 5? For instance, the authors may use boxplots to show the summaries for different regions mentioned in the study and in the figure. It may be helpful for exact explanations in the body text.

Version 1:

Reviewer comments:

Reviewer #1

(Remarks to the Author)

The authors have thoughtfully addressed my initial feedback. I appreciate both the responses and the edits to the manuscript. I am happy to recommend the revised draft for publication.

(Remarks on code availability)

Reviewer #3

(Remarks to the Author)

The manuscript has been critically improved. Thank you for your efforts in addressing my concerns. Your explanations are reasonable and revisions can clearly address the issues. I don't have further suggestions.

(Remarks on code availability)

The code is usable and the whole study is reproducible.

Reviewer #1 (Remarks to the Author):

“First, let me congratulate the authors on their research effort - this is a timely article. The manuscript examines infrastructure inequalities in light of urbanisation and economic develop using satellite remote sensing data in the form of NTL observations. This is an interesting endeavour, and an original contribution to the extent that it aims to address its main objective through four relevant research questions on infrastructure inequalities as a whole rather than some of its components. This is a well-written paper with a clear structure and sound internal consistency between theory and empirics. Therefore my review focuses on suggestions to improve it conceptually. I think the manuscript will be of immediate interest to people in the field of urban studies, urban sustainability, planning practitioners, and to those working on urban policy. In its present form, I would recommend authors to revise a few shortcomings before this manuscript can be accepted for publication.”

>>> Thank you for your constructive comments and feedback.

“First, in practice the paper deploys a narrow overview of sustainable development in the context of the relevance of the findings of you study. Here, I think your introduction would greatly benefit from a brief discussion on SDG11. The urban SDG is heavily based on indicators and urban data but this is not without critique. Perhaps you could engage theorists who have flag concerns with such approach. More specifically, there are arguments about the vagueness of the term “urban” as a definition that do not readily ‘fit’ into the urban focus of today’s global policy agendas (see Caprotti et al. 2017). Your article only briefly touches on this (lines 44-47). I encourage you to perhaps offer your definition or urban and what places were effectively considered urban before they were included in your analysis? Furthermore, since sustainability is holistically defined as development that advances human well-being equitably, a more thorough conceptual discussion of inequality would be warranted.”

>>> Thank you for pointing out the deficiency. To clarify, our conceptualization and measurement of infrastructure inequalities presented aligns with both the “degree of urbanization” conceptualization as well as the “planetary urbanization” framework.

In our study, we examined infrastructure inequalities across the complete rural-to-urban spectrum specified by variations within and across grid cells—we refer to each grid cell as an “urban region,” where each urban region can have a degree of urbanization. Capturing the complete rural-to-urban spectrum in-effect implements the “degree of urbanization” conceptualization recently brought to the fore by the United Nations¹, and addresses the point of neglected geographies raised in Caprotti et al.² . Additionally, we account for the diffusive nature of urban infrastructure and amenities extending from large urban areas to small settlements, aligning with the “planetary urbanization” framework³.

Our approach is developed to experimentally understand the association between infrastructure inequalities and urbanization/economic development, which necessitates capturing the complete rural-to-urban spectrum. In contrast, focusing only on urban areas, such as in SDG11, can leave out two important aspects:

- (1) the infrastructure inequality implications of urbanization can extend beyond administratively or physically defined urban areas; and**
- (2) the definition and identification of urban areas itself remain contentious.**

Lastly, our experimental design details outlined above indeed connects with the ideas that sustainable development, in part, is to advance human well-being equitably, which warrants examining the entire rural-to-urban spectrum. We have edited the introduction section to clarify the alignment of our approach with both the “degree of urbanization” conceptualization as well as the “planetary urbanization” framework. In light of our experimental design and findings, put in the context of SDG11, we clarify in the discussion section that addressing infrastructure inequalities vis-à-vis urbanization will require focus beyond urban areas.

“One limitation of analytical approach is the predictors of long-run infrastructure growth. As discussed by the authors, longer-run effects include crucial mechanisms that would affect infrastructure such as urbanisation rate and economic development, and quite possibly to larger extents. However, population dynamics seem to have been underrepresented, principally migration as a driver of population change. A typical example is internal migration, which in some rapidly urbanising cities in the Global South like Chattogram (formerly known as Chittagong), but many more examples in South and Southeast Asia and Latin America, have experience incredibly fast population growth mainly through net origin in-migration (see Mia et al 2015; Siddiqui et al 2021). How would one reconcile the long-term shifts in population growth with infrastructure projections? I am unsure as to whether this reconciliation, which directly impacts sustainable development - chief among which, the SDG 11 – but also inequalities within countries, can be incorporated in your forecast exercise. But I think Nature Comms readership would be interested to reading your thoughts.”

>>> We agree with the limitation of our current approach in not explicitly accounting for population migration. Here we provide a two-point clarification. First, even though inter- and intra-country migration can be significant driver of population changes, it was not the focus of our study. Instead, our approach utilizes spatially explicit population data, in addition to satellite nightlights-based infrastructure distributions, to generate inter- and intra-region inequality estimates. Since one of the outcomes of migration is changes in population distribution, our inequality estimation captures its impact—just not explicitly modeled. Second, long-term evolution of inequality and the impact of migration,

as a corollary, remains implicitly accounted for with a caveat that novel future migration patterns remain unaccounted for.

We believe that this is a significant and important area of research and warrants a future more elaborate effort, which can build on our approach and findings. We have added key points from our response to the discussion section.

“Your conclusion touches on an important aspect of urbanisation which was largely silent throughout the rest of the manuscript: that of socio-environmental implications. This point goes back to my first point on sustainable development. If the authors think that infrastructure inequality is somewhat a product of the interplay of demographic, economic, social, and environmental systems, then, in a way, only a simultaneous improvement on these four dimensions can move urbanisation towards sustainable outcomes. This intuitively makes sense for poorer countries, but in rich countries, is an increase in aggregate economic performance (a rise in GDPc) necessary to become more sustainable, and what effects increased sustainability bring to infrastructure inequality? I would argue that rich societies could become more sustainable if they managed to decrease their environmental burdens and existing social inequalities, which would then lead to more prosperous urbanisation. But this in turn raises another question: will efficiency gains in terms of improved sustainable outcomes for urbanisation globally be absorbed by capital owners and thereby contribute to enduring inequality, thus reinforcing civic infrastructural issues (see Sampson 2017).”

>>> Thank you for this insightful comment. Our key finding stresses urban infrastructure inequality as an inevitable, emergent feature of urbanization and economic development. It sets an expectation for rise in infrastructure inequality over time in urbanizing countries, locking in unequal infrastructure distributions as observed in more urbanized countries. Our finding asserts urbanization as an inherently unsustainable process—with a sustainable development definition to include the normative notion of reducing inequalities, such as in the UN SDG—and infrastructure inequality as a foundational constraint for sustainable development, challenging prescriptions that propose urbanization as a driver of sustainability.

Nevertheless, our finding has an important nuanced implication for global urban policy in that it emphasizes the criticality of prioritizing equity in infrastructure development vis-à-vis urbanization and economic development, for less and more developed countries. Our key finding is predicated on average relationships yet significant heterogeneity in the level of inequalities exist at a given level of urbanization and economic development suggesting the scope for planning more equitable infrastructure transitions. For instance, our results show heterogeneity in the level of inequalities for countries at similar levels of urbanization and economic development (see Figure 2 and Tables SI 1-4). Furthermore, infrastructure development is foundational to addressing many of the socio-

environmental challenges, across countries at varying levels of urbanization and economic development. Here, our results bring to the fore the challenge of spatial inequalities associated with infrastructure development.

Reviewer #2 (Remarks to the Author):

“This paper studies changing infrastructure inequalities from 2000 to 2019 using remote sensing. It finds the rise of infrastructure inequality in most of the countries, with more significant rise in the global south, especially the ones with more urban primacy. The paper also finds the positive associations between infrastructure inequalities and urbanization and economic development inequalities. The findings are interesting and confirm what we expect. So, what is new? Not much. What do positive associations between infrastructure inequalities and urbanization and economic development inequalities? Who causes whom? Are they bad? The paper raises the research gap that “our understanding of infrastructure inequalities accompanying urbanization is limited.” So, what is the contribution of this paper to the knowledge? The paper has not spelled it out, and I therefore am not sure about the contributions and implications. I was also trying to know the research progress in the field, but quite limited.”

>>> Thank you for your critical observations. We recognize that the previous version of the manuscript did not explicitly call out the novelty of our research and findings. Here, we offer three key clarifying points on the novelty of our study, as Reviewers #1 and #3 also note. First, to the best of our knowledge, our study is the first to empirically show widespread long-term increase in within and between region infrastructure inequality levels with rise in urbanization and economic development levels across countries. Second, our study is also the first to show this novel association using satellite remote sensing derived nighttime lights data to examine infrastructure distributions. Third, our study contributes to the existing literature with our infrastructure inequality forecasts that show significant future increases in infrastructure inequalities particularly in countries grappling with the urban primacy problem.

Infrastructure development has been frequently argued to be an essential driver of global sustainable development. However, our findings challenge such assertions owing to the attendant spatial inequality constraints our study highlights. Our findings emphasize the importance of an intentional approach, prioritizing infrastructure planning and development for spatial equity, as opposed to for aggregate development and treating inequality as an outcome of infrastructure development, which can be curbed by equity measures (e.g., Ramaswami et al. ⁴; ACERE ⁵; Bettencourt et al. ⁶), for all countries regardless of their urbanization and economic development levels, and with a spatial focus extending beyond urban areas. Our revised manuscript explicitly calls out these novel aspects of our research findings.

Reviewer #3 (Remarks to the Author):

“This study investigates the global rising infrastructure inequalities together with urbanisation and economic development using remote sensing data and future socioeconomic scenarios. From my perspective, the objective of this study is novel and ingenious. When we conducted studies about sustainable infrastructure development, it seemed that we made things complex as we spent too much time and effort on high-accuracy data collection of the infrastructure itself. This study provides a relatively simple but effective approach to assessing sustainable infrastructure, including developing relatively simple models to characterise inequality using NTL to quantify infrastructure, and predicting the future using socioeconomic scenarios. The structure and findings of the study are very clear. My suggestions for this study are more about the spatial data analysis methods and the results' explanations. My suggestion for the current manuscript is Major Revision.”

1. Line 53: The study addresses the listed four questions, but in the Abstract, the authors mentioned three findings from the study. Can you please consider making the questions and findings of the study consistent?

>>> We have edited abstract to make questions and our findings coherent.

2. Line 65: A few explanations should be added here to demonstrate that Earth observation, including remote sensing, has been an essential technology to monitor, assess, and manage sustainable infrastructure development in previous studies. Earth observation for sustainable infrastructure critically contributes to implementing Earth data and infrastructure sustainability.

>>> We added a brief discussion of using Earth-observation data for assessing sustainable infrastructure development, also in response to Reviewer #2's comment.

3. Line 139 and Fig 1: The study area is firstly divided into multiple regions and the regional differences are compared to calculate the inequality. However, the approach of dividing regions has critical impacts on the quantification of inequality, which has been proved in a series of studies about spatial data discretisation and spatial stratified heterogeneity, including doi: 10.1080/15481603.2020.1760434 and many relevant studies. Methods of spatial discretisation generally include many types of supervised and unsupervised approaches. The common approach of spatial discretisation is to determine the optimal combinations of numbers and methods of discretisation that can derive the maximum spatial heterogeneity (OPGD), i.e., inequality in this study. Optionally, authors may also consider a few advanced models for spatial discretisation, such as GOZH and RGD. This means that the inequality calculated in the study will significantly change if you use different methods to divide the study area. I highly recommend authors select a relatively reasonable approach, at least the typical approach of spatial discretisation, to divide the study area for computing inequalities. I believe if you choose other spatial discretisation methods, the study's results will be changed.

>>> Thank you for this constructive comment. We agree that the level of inequality changes with how the study area of interest is divided. To control this effect, however, was the prime reason that our research design uses a uniform grid—where each grid cell denotes a region.

In the early phases of our research, we experimented with spatial discretization methods to estimate infrastructure inequalities. However, we found two issues with such

approaches for understanding changes in inequality. First, spatial discretization methods maximize between cluster/region variance and with infrastructure development, such methods are likely to suggest greater changes in between-region inequality than within-region inequality. Second, differences in region demarcation between two countries itself becomes a driver of measured inequality. To overcome these issues, we decided to use the uniform grid.

Next, as shown in our previous study (Pandey et al., 2022), grid resolution can influence inequality levels. Thus, in our revised manuscript, we have added supplementary figures showing that our key finding is robust to changes in grid resolution. Specifically, we show a positive association between within-/between-region inequality and urbanization/economic development levels across countries. Similarly, the widespread increase in within-/between-region inequality across a preponderance of countries is observed even at varying spatial scales, specified by grid resolutions (see Figures 2 and 3).

4. Line 318: The study investigates infrastructure in urban regions. How did you define urban regions? Which data set of urban boundaries do you use to define urban areas? In the figures, I assume that the authors include both urban and the surrounding regions, which may be rural or non-residential areas, in computing inequality. The selection of rural and non-residential regions may critically impact quantifying inequality.

>>> Thank you for pointing out the lack of clarity. As also part of our response to Reviewer #1, we have edited the manuscript to improve clarify by incorporating key points from the following response:

“Our conceptualization and measurement of infrastructure inequalities presented aligns with both the “degree of urbanization” conceptualization as well as the “planetary urbanization” framework. In our study, we examined infrastructure inequalities across the complete rural-to-urban spectrum specified by variations within and across grid cells—we refer to each grid cell as an “urban region,” where each urban region can have a degree of urbanization. Capturing the complete rural-to-urban spectrum in-effect implements the “degree of urbanization” conceptualization recently brought to the fore by the United Nations. Additionally, we account for the diffusive nature of urban infrastructure and amenities extending from large urban areas to small settlements, aligning with the “planetary urbanization” framework. Our approach is developed to experimentally understand the association between infrastructure inequalities and urbanization/economic development, which necessitates capturing the complete rural-to-urban spectrum.”

5. Line 450: The authors mentioned that population is closely linked with urbanisation. This study analyses infrastructure inequality based on spatial heterogeneity and from a spatial perspective. Can you please explain the inequality of the population's share of infrastructure, i.e., infrastructure per person?

>>> Thank you for this insightful comment. Here, we clarify that our historical infrastructure inequalities measurement indeed accounts for population distribution, as a population weighted measure.

6. Fig 5: Can you please consider adding a figure to the statistical summary of findings from Fig 5? For instance, the authors may use boxplots to show the summaries for different regions mentioned in the study and in the figure. It may be helpful for exact explanations in the body text.

>>> Thank you for this suggestion. We have edited Figure 5 to include LOESS fit line, highlighting no relationship between income and infrastructure inequality forecasts across all countries.

Reviewer #1 (Remarks to the Author):

“The authors have thoughtfully addressed my initial feedback. I appreciate both the responses and the edits to the manuscript. I am happy to recommend the revised draft for publication.”

>>> We thank the reviewer for their constructive comments and feedback.

Reviewer #3 (Remarks to the Author):

“The manuscript has been critically improved. Thank you for your efforts in addressing my concerns. Your explanations are reasonable and revisions can clearly address the issues. I don't have further suggestions.”

>>> We thank the reviewer for their valuable comments and feedback.

Reviewer #3 (Remarks on code availability):

“The code is usable and the whole study is reproducible.”

>>> We thank the reviewer for reviewing our code.